# Neutrophils in the Spotlight—An Analysis of Neutrophil Function and Phenotype in ARDS

**DOI:** 10.3390/ijms252312547

**Published:** 2024-11-22

**Authors:** Richard F. Kraus, Lisa Ott, Kirsten Utpatel, Martin G. Kees, Michael A. Gruber, Diane Bitzinger

**Affiliations:** 1Department of Anesthesiology, University Hospital Regensburg, Franz-Josef-Strauss-Allee 11, 93053 Regensburg, Germany; 2Institute of Pathology, University of Regensburg, Franz-Josef-Strauss-Allee 11, 93053 Regensburg, Germany

**Keywords:** ARDS, neutrophil, surface epitope, ROS, intensive care, flow cytometry

## Abstract

Acute respiratory distress syndrome (ARDS) is a complex disease pattern in which pathogenesis polymorphonuclear neutrophil granulocytes (PMN) play a key role. In previous experiments, we could show that interaction with collagen III (an important component of pulmonary tissue) is a possible trigger of neutrophil reactive oxygen species (ROS) production. To investigate possible correlations, further elucidate ARDS pathophysiology, and maybe find pharmacological targets, we evaluated PMNs from blood (circulating PMNs: cPMNs) and tracheal secretion (tPMNs) from patients with and without ARDS with regard to function and phenotype. Blood samples and tracheal secretions were obtained from intensive care patients with and without ARDS. Isolation of cPMN was performed by density-gradient gravity sedimentation without centrifugation. For tPMN isolation, endotracheal aspirate was filtered, and tPMNs were separated from the remaining aspirate using a particle filter. Specific surface epitopes (CD66b, CD62L, fMLP-receptor, LOX-1, CD49d, CD29, CD11b) of the isolated PMN cells were labeled with antibody-coupled dyes and analyzed by flow cytometry. Neutrophil ROS production before and after activation with N-formyl-methyl-leucyl-phenylalanine (fMLP) and tumor necrosis factor α (TNFα) was quantified using rhodamine-123. In addition, a qualitative cytological hematoxylin-eosin (HE) staining was performed with a portion of the secretion. tPMNs were observed in both bloody and mucosal tracheal secretions from ARDS patients. The epitope distribution on cPMNs and tPMNs differed significantly in patients with and without ARDS: tPMNs generally showed increased expression of CD66b, LOX-1 and fMLP-receptor compared to cPMNs, and decreased expression of CD62L. The CD49d levels of all cPMNs were at the same level as tPMNs in ARDS, whereas CD49d expression was increased on tPMNs without ARDS. ROS production was significantly stimulated by fMLP/TNFα in cPMNs regardless of the patient group, while it was similarly increased in tPMNs with and without stimulation. Increased expression of CD66b, LOX-1 and fMLP-receptor on tPMNs indicated a higher activity status compared to cPMNs. Increased CD49d expression on tPMNs without ARDS marks different PMN surface changes in lung disease. PMNs appear to be in a more activated state in lung secretions than in blood, as indicated by higher CD66b and lower CD62L expression, higher constitutive ROS production and lower excitability with fMLP and TNFα. In the context of possible CD49d-triggered ROS production, it is noteworthy that CD49d is downregulated in secretion from patients with ARDS compared to patients without. This phenotypic and functional PMN characterization can provide valuable diagnostic and therapeutic information for the intensive care treatment of ARDS patients.

## 1. Introduction

### 1.1. Neutrophil Involvement in the Pathophysiology of Acute Respiratory Distress

ARDS is a clinical disease pattern associated with diffuse pulmonary inflammation and edema leading to hypoxemia. Global awareness of ARDS was heightened by the COVID-19 pandemic in 2020, which resulted in a sharp rise in ARDS incidence. ARDS can be triggered by a variety of infectious or non-infectious causes, such as pneumonia or extra-pulmonary sepsis, pancreatitis, aspiration of gastric contents or severe trauma with shock and multiple blood transfusions [1,2,3,4].

The pathophysiology of ARDS involves the activation and dysregulation of multiple overlapping and interacting pathways of injury, inflammation and coagulation, both locally in the lung and systemically [5,6]. Polymorphnuclear neutrophils (PMNs) are significantly involved in these processes [7]. PMN-triggered damage to the alveolar-capillary barrier plays a key role in the pathophysiology of ARDS. The upregulation of adhesion molecules such as P-selectin (CD62P) and E-selectin (CD62E) leads to the migration of inflammatory cells [8,9]. PMNs migrate along the chemotactic gradient into the lung parenchyma. Reactive oxygen species (ROS) production, degranulation and Neutrophil Extracellular Trap (NET) formation in an injured alveolus result in damage to the alveolar epithelium, which attracts further immune cells and leads to endothelial dysfunction [10,11,12,13]. Neutrophilic alveolitis and hyaline, fibrin- and type III collagen-rich and proteinaceous membrane deposits ultimately lead to diffuse alveolar damage. Severe damage to the lung epithelium can also lead to damaged lung endothelium [2].

Activation of procoagulatory pathways on pulmonary endothelium and hypoxic vasoconstriction can lead to microvascular thrombosis in the lung, in the development of which PMNs are involved. As a result, the dead space increases and gas exchange is further impaired [2,14,15].

The success of drug-based treatment of ARDS varies. Salton et al. found that while glucocorticoids reduce mortality and the need for invasive mechanical ventilation in SARS-CoV-2-induced ARDS, patients requiring intubation exhibited higher inflammation levels, suggesting potential resistance to glucocorticoid treatment [16]. Especially in COVID-19 patients, elevated serum levels of eotaxin, type 1 and type 2 cytokines, and alarmin cytokines indicate a heightened immune response, while lower levels of eosinophils and their degranulation products, combined with reduced eosinophil activity, underscore the complex interplay of immune mediators in the pathogenesis of ARDS [17]. Although there are new therapeutic targets like the mammalian enzyme neuraminidase-1, which are predicted to be highly effective against infection with SARS-CoV-2, cellular infectivity and the induction of the cytokine release syndrome and despite intensive research to date, the exact pathophysiological involvement of PMNs in the clinical picture of ARDS is still not fully understood [7,10,18].

### 1.2. Selection of the Investigated Surface Epitopes

To further elucidate the involvement of neutrophils in ARDS pathophysiology, certain neutrophil surface epitopes possibly important for pathophysiology were investigated in the context of ARDS. The surface epitopes studied were selected on the basis of their function in the neutrophil immune response. Neutrophil recruitment to inflammatory pulmonary tissue requires adhesion to and subsequent transmigration through the vascular walls of pulmonary capillaries [19]. The initial action of PMN extravasation is activation and upregulation of adhesion molecules: CD62L is one of the first neutrophil adhesion molecules to be in contact with the endothelium under flow conditions in the bloodstream. CD62L is essential for transmigration through blood vessels and, therefore, upregulated in inflammatory conditions. CD11b upregulation enables activated PMNs to adhere to vascular endothelium and initiates the transmigration [19]. CD66b is a granulocytic-specific receptor that mediates phagocytosis [20]. Originally described as a receptor for oxidized low-density lipoprotein, LOX-1 was identified as an adhesion molecule for PMNs and is now recognized as a multi-ligand receptor that acts in a compendium of different cellular processes [21,22]. The fMLP receptor is G-protein coupled and mediates inflammatory reactions in human neutrophils [22]. PMNs express integrins with α4-subunit (such as α4β1 (CD49d/CD29)), which are related to the collagen receptor integrin family. α4-integrins are able to recognize extracellular matrix proteins such as pulmonary type III collagen, whereby a direct cross-linking of α4 integrins triggers the release of superoxide [23,24,25]. In a previous study, we recently observed increased neutrophil ROS production when PMNs have contact with type III collagen [26]. We observed type III collagen to have an inhibitory effect on PMN migration regarding track length, direction, and targeting, as well as an accelerating effect on neutrophil ROS production. It is striking that pulmonary tissue is rich in type III collagen, which is the main component of the fibrils of the alveolar walls and septa, and that PMNs are significantly involved in the pathology of ARDS [26]. Moreover, as described by Pugin et al., intensive inflammation and increased synthesis of type III collagen already occur at the alveoli in the early phase of ARDS [27]. We hypothesized that integrins with an α4 subunit (CD49d) may trigger neutrophil ROS production via direct cross-linking of α4 integrins [24,26,28].

In this way, the investigation of expression levels of these surface epitopes not only enables further elucidation of ARDS pathophysiology, but may also help to discover targets for new pharmacological therapies. To investigate a possible connection in more detail, we carried out cytological examinations and immunophenotyping in patients with and without ARDS and also examined the ROS production of PMNs from blood and tracheal secretions.

## 2. Results

### 2.1. Demographic Data of the Included Patients

A broad population structure was included in the study (see Table 1 and Table 2). More information about the included patients is attached in Appendix A.

### 2.2. Macroscopic Properties of the Endotracheal Aspirate

Macroscopically, clear differences between obtained samples could be determined (see). Colorless, clear (Figure 1a), greenish-mucous (Figure 1b) and red-sanguineous (Figure 1c) samples were observed. In the ARDS group, all variants occurred; without ARDS, only serous aspirate was observed.

### 2.3. Cytological Staining of the Endotracheal Aspirate

In both mucosal (Figure 2a–c) and sanguineous (Figure 2d,e) tracheal aspirates, massive immigration of PMNs into the mucus was detected. In serous aspirates, only a few PMNs could be observed.

### 2.4. Cell Counts in Endotracheal Aspirate

The mean cell number in the endotracheal aspirate (*n* = 10) determined by the CASY cell counter was 262 $\frac{Cells}{\mu L}$. The comparison with cell numbers determined by the flow cytometry (*n* = 6 measured tubes and *n* = 3 different samples) yielded 11 cells per µL. An average of 7736 cells were obtained from one sample, which was divided into 3 flow cytometry tubes.

### 2.5. Results of Flow Cytometry Measurements

The results of the flow cytometric measurements are shown in Figure 3 and described in the sections below.

#### 2.5.1. Results of Dead or Alive Testing

The percentage of avital PMN cells was significantly lower for cPMNs than for tPMNs (Figure 3a and Appendix A).

#### 2.5.2. Results of CD11b, CD66b and CD2L Expression on PMN Surfaces

Analysis of CD11b revealed no significant differences (*p* = 0.718) between cPMN and tPMN or between groups. The results of median values CD11b are listed in Appendix A. CD66b was significantly less expressed on cPMN without ARDS than in all other three categories (see Figure 3b and Appendix A). The expression of CD66b was higher and more variable in tPMN than in cPMN and lower in cPMN of patients with ARDS than without. CD62L was significantly highest on cPMN without ARDS compared to all other groups (see Figure 3c and Appendix A).

#### 2.5.3. Results of LOX-1 Expression on PMN Surfaces

LOX-1 of cPMN without ARDS was significantly lower than of tPMNs with and without ARDS. Similarly, LOX-1 of cPMN with ARDS was significantly lower than tPMN with and without ARDS (see Figure 3d and Appendix A).

#### 2.5.4. Results of CD49d/CD29 Expression on PMN Surfaces

CD49d of tPMN without ARDS was significantly higher (see Figure 3e and Appendix A). Similarly, CD29 of cPMN without ARDS was highest compared to the other groups, but post hoc analysis did not reveal any significant differences (see Appendix A).

#### 2.5.5. Results of fMLP-Receptor Expression on PMN Surfaces

FMLP-receptor expression was lower in cPMNs compared to tPMNs regardless of whether patients had or did not have ARDS. There were no differences between cPMNs from patients with or without ARDS in fMLP-receptor expression. Similarly, there were no significant differences between tPMNs from patients with or without ARDS (Figure 3f and Appendix A).

### 2.6. Results of the ROS Measurements

Analysis of the fluorescence intensity of Rhodamine-123 without stimulation revealed significant differences between the four groups (see Figure 4).

The blank value of Rhodamine-123 of cPMN without ARDS was significantly lower than Rhodamine-123-fluorescence of tPMN with and without ARDS, indicating less basal activity. Similarly, Rhodamine-123 of cPMN with ARDS was significantly lower than that of tPMN with and without ARDS (see Appendix A).

After stimulation with FMLP + TNFα, Rhodamine-123 of cPMN without ARDS was significantly higher than that of tPMN with and without ARDS, indicating higher excitability. Similarly, Rhodamine-123 of cPMN with ARDS was significantly higher than that of tPMN with ARDS (see Appendix A).

After stimulation with PMA, Rhodamine-123 of cPMN without ARDS was significantly higher than that of tPMN with and without ARDS. Similarly, Rhodamine-123 of cPMN with ARDS was significantly higher than that of tPMN with and without ARDS, indicating higher excitability (see Appendix A).

## 3. Discussion

### 3.1. Interpretation of the Macroscopic Properties and Cytological Staining of the Endotracheal Aspirate

Macroscopic assessment of the endotracheal aspirate provides information on the physiological state of the airways and allows conclusions about potential changes in the course of ventilator-dependent lung diseases. Serous transparency of the aspirate (Figure 1a) shows no admixture of blood or mucus. These clear characteristics indicate a non-pathological composition of the aspirate with a higher liquid content and a low solid content. Since this appearance was particularly present in the control group without ARDS, it can serve as a reference point for comparison with the other macroscopic appearances “mucous” (Figure 1b) and “sanguineous” (Figure 1c) in the presence of ARDS. Serous aspirate indicates a “more normal” secretion and absence of pathological changes [29].

If the consistency of the aspirate shows viscous mucus formation (Figure 1b), this indicates a possible increase in mucus production in the airways. Inflammatory processes activate the mucus-producing cells to various stages [30]. In ARDS, there is an intraluminal accumulation of PMNs in the airways and mucus. This visual representation enables an assessment of the amount and viscosity of the mucus, which in turn provides important information about the condition of the airways, possible infections or inflammatory processes [29].

Sanguineous staining of the aspirate (Figure 1c) indicates blood admixture, which may be related to vascular injury, hemorrhagic events or injury to the capillary alveolar membrane [11], which is typical in the course of ARDS [27]. Sanguineous samples were obtained, particularly in the late phase of ARDS. The cytological staining revealed that PMNs could be detected in the intratracheal secretion in all manifestations (serous, mucous, sanguineous). In terms of optical quantity, more PMNs were observed in mucous and sanguineous secretions. This can be interpreted as an expression that the more severe the ARDS, the more PMNs are found endotracheally (and ultimately also intrapulmonary) [10].

### 3.2. Results of Surface Epitope Distribution with and Without ARDS

The analysis of surface epitopes showed striking differences in PMNs of blood and endotracheal aspirate between patients with and without ARDS for all examined groups.

#### 3.2.1. Interpretation of CD11b Expression in ARDS

PMN priming is partly mediated by the mobilization of intracellular granules, which leads to a reorganization of surface epitopes. CD11b is upregulated by this release of granules and secretory vesicles [31].

Inflamed lungs show increased VCAM-1 expression. Expressed on PMNs, CD11b binds to the cell adhesion molecule VCAM-1 as a receptor component so that the cells adhere and migrate through the endothelium to the site of inflammation [32,33]. ARDS is associated with increased CD11b/CD18 expression on PMN cell surfaces, so changes in circulating blood PMNs may be markers for the development of ARDS [34].

We did not detect significant differences in CD11b between cPMNs and tPMNs, indicating that tPMNs (are still able to) express CD11b. In contrast to our data, Chollet-Martin et al. found increased CD11b levels in BALs of ARDS patients compared to blood [35]. Elevated CD11b on tPMNs with ARDS compared to tPMNs without ARDS indicates that CD11b in the lung may have additional functions in ARDS. Therefore, from an immunological point of view, CD11b is essential in both blood and endotracheal aspirate to ensure the integrity of the innate immune system [36].

#### 3.2.2. Interpretation of CD62L Expression in ARDS

While CD11b is upregulated by the release of granules and secretory vesicles, the cell adhesion molecule CD62L is cleaved from the neutrophil surface by proteolysis [31,37]. An increased surface density of CD11b and a lower concentration of CD62L are, therefore, a typical pattern for PMNs that have been primed [38].

Lower CD62L values on tPMN with and without ARDS in our study could be explained by the diapedesis and the priming that had already taken place at this point. In accordance, Chollet-Martin et al. showed with samples from ARDS patients that CD62L was lower in the PMNs from BALs compared to the corresponding blood PMNs [35]. PMNs in the lumen of the airways, therefore, exhibit higher priming than circulating PMNs in the blood. Nevertheless, they still have the potential to be further primed by local cytokines [38]. Priming of the PMNs could protect against ARDS or, on the contrary, even precede ARDS and, through a type of excessive reaction of the PMNs, favor development or even cause ARDS. The fact that CD62L was already lower in cPMN with ARDS than in cPMN without ARDS already indicates initial priming in the blood of ARDS patients.

#### 3.2.3. Interpretation of CD66b Expression in ARDS

CD66b expression on neutrophil surfaces increases after stimulation. It performs chemotaxis functions (cell adhesion, cell migration) and also binds pathogens [39,40]. Therefore, the massively increased expression in cPMN with ARDS in our study can be explained by an increased activation of PMNs. The binding of pathogens as an elementary function of the molecule could also explain its expression on tPMN without ARDS. In the blood, the role of a cell adhesion and cell migration mediator could predominate.

CD66b is granulocyte-specific and may, therefore, be important as a biomarker in ARDS, as it indicates, for example, an extracellular vesicle originating from PMNs. Extracellular vesicles represent an important intercellular communication mechanism that enables the targeted transfer of biological cargo such as RNA, micro-RNA, proteins and mitochondria between different cell types [41,42].

Our results show elevated CD66b levels on tPMN, which suggests that similar mechanisms may be present in patients who develop ARDS after similar remote (indirect) insults. This mechanism could partly explain the development of neurogenic pulmonary edema and lung injury after shock and ischemia-reperfusion [41,42].

Epithelial extracellular vesicles are enriched with tissue factors and likely contribute to increased pro-coagulatory activity. Increased concentrations of endothelial extracellular vesicles are also associated with increased mortality in ARDS patients [42,43]. Moreover, PMNs expressing the dual endothelin1/signal peptide receptor in combination with CD11b and CD66b (DEspR+CD11b+/CD66b+) are associated with severity of hypoxemia and multi-organ failure in ARDS and might provide an actionable therapeutic target in ARDS [44].

#### 3.2.4. Interpretation of LOX-1 Expression in ARDS

During transmigration from the bloodstream into the lung through the endothelial-epithelial bilayer, PMNs are exposed to a new environment and change their function on the way there or through transmigration [45].

The results of our study showed a significantly higher expression of LOX-1 on tPMNs compared to cPMNs in healthy individuals and in ARDS. To the best of our knowledge, we were the first to examine LOX-1 levels on PMNs in the context of ARDS. Their activation state seems to change, which is reflected in their altered surface epitope LOX-1. Even in healthy individuals, LOX-1 may be important for the maintenance of physiological PMN activity in the lungs.

Interestingly, the scavenger receptor LOX-1 is primarily known to promote endothelial dysfunction by triggering proatherogenic signaling and plaque formation through the uptake of oxidized and electronegative LDL into endothelial cells [21,46]. LOX-1 is, therefore, upregulated after exposure to various proinflammatory and proatherogenic stimuli [47]. PMNs from patients with severe COVID-19 disease and sepsis showed extremely high expression of LOX-1 [48]. In infected lungs, PMNs show a rapid but heterogeneous increase in LOX-1 [49]. It is, therefore, hypothesized that recruited neutrophils are an important site of LOX-1 expression in the lung, suggesting that intrapulmonary LOX-1 activity plays a role not only in acute lower respiratory tract infections but also in lung injury [49]. For example, a study on COVID-19 showed that LOX-1 is overrepresented in intensive-care patients compared to non-intensive-care patients and that the proportion of LOX-1-expressing PMNs is positively correlated with the clinical severity of the disease. Thus, the higher the LOX-1 expression, the stronger the cytokine storm by IL-1β, IL-6, IL-8 and TNFα, and the more severe the ARDS and thrombotic complications. The BALs of patients with ARDS were highly enriched with LOX-1-expressing PMNs [50].

LOX-1, for example, has a protective function in the alveoli by curbing protein edema and inflammation. Air spaces in the lung appear to provide a niche for LOX-1-driven tissue protection, possibly by regulating the activity of alveolar macrophages and recruited PMNs to maintain tissue homeostasis after infection [49]. This is a possible explanation for high values in the endotracheal aspirate compared to blood.

#### 3.2.5. Interpretation of fMLP-Receptor Expression in ARDS

In general, activation of the fMLP-receptor leads to neutrophil activation. Increased pathogen exposure in the lungs in conjunction with high fMLP-R values (observed in our study) indicates an increased activation state of the tPMNs in comparison to cPMNs. Physiologically, PMNs recognize chemokines, including the bacterial product FMLP, complement C5a or lipid molecules such as leukotriene B4 (LTB4) and migrate to the intended site of inflammation. In septic patients, PMNs may not migrate to the correct target site despite very high chemokine levels but instead accumulate in the lungs, where they release histones, DNA and proteases, which mediate further tissue damage and trigger ARDS.

Dysregulation of chemokine receptors may be an important cause of this impaired migration, and the vast majority of these chemokine receptors are G protein-coupled receptors. However, our results showed no differences in expression between tPMN with or without ARDS, indicating a subordinate role of this receptor in ARDS. Nevertheless, It cannot be ruled out that the function and sensitivity of the neutrophil fMLP-R may be impaired in ARDS [10,51].

#### 3.2.6. Interpretation of CD49d/CD29 Expression in ARDS

If pulmonary inflammation occurs, CD49d is probably involved in the diapedesis of mononuclear cells into the inflamed lung, as shown by Yen, Liao et al. in the context of SARS-CoV-2 infection [28]. With regard to viruses, integrins probably also have an inhibitory role against virus entry into the cells [52,53].

In our experiments, tPMN without ARDS showed significantly higher levels, which is compatible with an inhibitory role with regard to virus entry into lung cells under physiological conditions. In addition, CD49d could be responsible for attracting PMNs from the blood, where low levels are present, into the lungs. In ARDS, a reduced expression of CD49d in the secretion could be involved in the development due to a lack of viral inhibitory effects.

In addition, CD49d mediates both rolling and firm adhesion [54,55]. Reduced CD49d expression on PMNs in the endotracheal aspirate of ARDS patients could indicate impaired pulmonary PMN migration due to reduced adhesion in this condition. Since the role of the molecule in the development of other diseases has been established, it could also represent a drug target.

In combination with CD49d, the integrin α4β1, also known as VLA-4 (CD49d/CD29), is obtained [56,57]. The possible linking of CD29 and CD49d to a common receptor explains the similar distribution patterns of CD49d and CD29 in all four groups. CD29 assumes the function of a collagen receptor. This could explain the low levels of tPMN with secretion in our experiments compared to the tPMNs without ARDS: collagen receptors are membrane proteins that bind the extracellular matrix protein collagen, the most abundant protein in mammals. They primarily control cell proliferation, migration and adhesion, as well as the activation of the coagulation cascade. In addition, the extracellular matrix structure is influenced by the regulation of the matrix [58,59]. The reduced CD29 function due to lack of expression could play a role in the pathophysiology of ARDS (Figure 5). This is because changes in cell proliferation (fibrosis), migration and adhesion (neutrophil invasion) and activation of the coagulation cascade (thrombus formation) occur in ARDS [2,12,14].

Altered levels of CD49d/CD29 integrins, which interact with type III collagen, alongside increased ROS activity during the interaction of PMNs with type III collagen, indicate a significant association. Modulating this interaction may provide the potential for a pharmacological target for possibly alleviating severe ARDS [60].

### 3.3. Result Interpretation of ROS Measurements

Our results showed an increased intrinsic ROS production in tPMN with ARDS without stimulation. This suggests that the oxidative metabolism of PMN increases at the onset of ARDS [34].

With regard to the pathophysiology of ARDS, there is a theory that delayed neutrophil apoptosis is responsible for disease development. PMN migration through the pulmonary endothelial-epithelial bilayer suppresses PMN apoptosis. The prolonged lifetime and activity of PMNs in the inflammatory lung then leads to tissue damage [45]. Reduced activity and/or expression of proapoptotic proteins may represent candidates for regulating inappropriate delay of PMN apoptosis during pneumonia and ARDS [45]. In addition, the severity of septic ARDS correlates positively with the degree of neutrophil infiltration and the intensity of proteolytic enzymes derived from them in the BAL [61]. Our experiments showed that PMNs in ARDS could still be stimulated (by TNF/fMLP or PMA) despite high activation in ARDS but produce less ROS, which is consistent with observations available in the literature to date: PMNs from patients with severe COVID-19 disease and sepsis had a defective oxidative burst and phagocytosis despite extremely high expression of cellular and soluble activation markers (such as LOX-1) [48]. This reinforces the findings above with respect to the LOX-1 expression that tPMNs already seem to be in an activated state that cannot be augmented to a large extent. cPMNs, on the other hand, can be activated regardless of the condition with or without ARDS [62]. Maybe tPMNs are already in a later stage of their life cycle compared with cPMNs, whereby they are still capable of performing innate immune functions, which indicates their functional potential and contribution to a healthy pulmonary ecosystem [63]. However, direct modulation of pulmonary PMN cell death may represent a novel mechanism to attenuate PMN-mediated lung injury and may ultimately improve outcomes in patients with ARDS [64].

### 3.4. Features and Limitations of the Selected Methods

#### 3.4.1. Cell Count as Parameter for Process Optimization

A decisive step in result validation was made by comparing the results from the CASY cell counter with the cell counts determined on the FACS after filtering and processing the samples (see Section 4.4). Lower cell numbers in endotracheal aspirate compared to unfiltered endotracheal aspirate illustrate that cells are “lost” in the used filtering and isolation methods so that they were not retained after the first filter step or remained in the filter during backwashing. The cell number would have been, therefore, reduced by our filtering methods to 1/25 and needed to be improved (see Section 4.4). Inclusion of all particles > 8 µm was aimed to detect neutrophils in particular, but also monocytes and lymphocytes, and preferably no erythrocytes. However, fluctuations in the cell counts can occur between individual samples. These were presumably dependent on the consistency, composition and total volume of the sample.

#### 3.4.2. Improving Process Quality for PMN Preparation

Validation of the measured cell counts by flow cytometry and CASY measurements made it possible to show cell losses by the filtering steps and helped to verify cell losses by filtering methods used (Table 3). One applied improvement was to use filtering sequences that were as restrictive as possible, provided that the consistency of the samples was already homogeneous. Filtering with a 100 µm filter was completely dispensed with serous samples, while homogenization was instead achieved for mucous samples by gentle pipetting up and down.

#### 3.4.3. Cell Count Control as a Quality Feature

Following the considerations discussed in Section 3.4.1 and Section 3.4.2, measured cell counts of each flow cytometric tube were checked as a practical consequence and documented in connection with the corresponding experiment. The results of the evaluation were then subjected to a plausibility check. Experiments with implausible values, for example, fluorescence intensities < 0, could thus be explained by the low cell count.

## 4. Materials and Methods

Figure 6 provides an overview of flow cytometric experiments conducted in this study.

### 4.1. Vote of the Ethics Committee

The conduct of this study was approved by the local Ethics Committee of the Medical Faculty of the University of Regensburg (file number 23-3432-1-101).

### 4.2. Selection of the Patients

A total of *n* = 10 patients with diagnosed ARDS and *n* = 12 patients without ARDS were included in this study as a control group. All ARDS patients were in an intensive care unit at Regensburg University Hospital and ventilated via an endotracheal tube or tracheal cannula. The diagnosis of ARDS was made by experienced intensive care physicians according to the Berlin definition [4]. Subcategorization into different ARDS severity degrees was not performed. The patients in the control group were without ARDS and underwent elective surgery under general anesthesia with endotracheal intubation.

### 4.3. Sample Collection

For the isolation of cPMNs, whole arterial blood samples were taken from patients of the two defined groups into a lithium heparin blood collection tube (blood gas monovette 1 mL, Sarstedt, Nümbrecht, Germany). Herein, only residual blood was used, which was obtained as part of routine blood gas analysis measurements. The samples of endotracheal aspirate were obtained when endotracheal suctioning was required according to the clinical judgment of the treating physicians or nurses. The endotracheal aspirate was aspirated by negative pressure through an endotracheal tube or tracheal cannula by negative pressure using a suction catheter and collected in a tracheal suction set (SMS medipool GmbH, Germering, Germany (Figure 7).

### 4.4. Sample Preparation and Cell Isolation

One blood sample and one secretion sample were used per patient to conduct the experiment. For cPMN isolation, the blood samples were first mixed with gelafundin^®^ (ISO 40 mg/mL, B. Braun SE, Melsungen, Germany) as described by Hundhammer et al. [65]. After 30 min, sedimentation produced a supernatant containing the cells to be analyzed. The preparation of the endotracheal aspirate always followed the same basic scheme and was adapted to physiological differences in homogeneity and viscosity fluctuations (high viscosity vs. serous) of the samples (Figure 8). Highly viscous respective inhomogeneous samples were first liquefied with 0.9% NaCl solution and homogenized by careful pipetting up and down. Then, the samples were filtered through a 100 µm filter (pluriStrainer 100 µm, pluriSelect Life Science, Leipzig, Germany). Afterward, the tPMNs were then carefully separated from the medium without centrifugation using a Whatman syringe filter (Whatman Puradisc 13, pore size 5.0 µm, Cytiva, Marlborough, Massachusetts). PMNs with a diameter above 8 µm, which therefore initially remained in the filter, were rinsed back into a new container (tube 5 mL PP (FACS tube) Sarstedt AG and Co., Nümbrecht, Germany) in a second step after turning the filter by 180° and attaching a three-way stopcock (B. Braun Discofix^®^-3 three-way stopcock 360° rotatable, Braun, Melsungen, Germany). This backwashing of the PMNs from the syringe filter was performed with PBS buffer (Dulbecco’s Phosphate Buffered Saline with MgCl_2_ and CaCl_2_, liquid, sterile-filtered, suitable for cell culture (D8662-500ML) Sigma-Aldrich, Steinheim, Germany), resulting in a total volume of 700 µL secretion cell solution. For serous samples, these filtering steps were omitted so that the clear aspirate was immediately diluted to 700 µL as a secretory cell solution. The cells from this solution were then used as secretion samples for all experiments.

### 4.5. Flow Cytometric Analysis of Surface Epitopes and ROS Production of PMNs

A separate analysis of neutrophil ROS production and surface epitopes (Figure 9) was performed per patient by flow cytometry.

### 4.6. Experimental Procedure Surface Epitope Analysis

The isolated cells were labeled with fluorescence-labeled antibodies to analyze neutrophil surface epitopes (Table 4).

For antibody binding, 20 µL PMNs from the blood sedimentation were pipetted into 4 tubes each (tube 5 mL PP (FACS tube) Sarstedt) and increased to a volume of 70 µL by adding 50 µL PBS buffer (D8662-500 µL). The PMNs from the secretion were divided equally into four measuring tubes with 100 µL each. Antibodies from the blood and secretions were not added to the first tubes of PMNs in order to generate a blank value for later evaluation. In the other three tubes of both preparations, 5 µL of the antibodies were added for incubation. The second tube was incubated with the antibodies against CD11b, CD62L and CD66b, the third with the antibody against LOX-1 and the fourth with the antibodies against CD49d, CD29 and the fMLP receptor. Antibodies that were processed in one tube were coupled to different dyes. Antibodies that were coupled to the same dye were distributed to different tubes. The apportionment of the antibodies is shown in Table 5.

### 4.7. Experimental Procedure for PMN ROS Production

Neutrophil ROS production was examined by flow cytometry parallel to surface epitope analysis. Dihydrorhodamine-123 (DHR, Thermo Fisher Scientific) and SNARF (Carboxy SNARF, Thermo Fisher Scientific) were used to visualize ROS (see Table 6).

After 20 µL blood cells or 100 µL secretory cells were placed in each of the three tubes, 5 µL DHR and 5 µL SNARF were added. As in the experiment for surface epitope analysis, the cells in the first tube were used as a blank value and, unlike tubes two and three, were not treated with activating substances. The cells in the second tube were primed with 5 µL tumor necrosis factor-alpha (TNFα) (Human TNF Perotech 1 µg/mL, PeproTech Germany, Hamburg, Germany) [66]. The PMNs were then activated using 5 µL fMLP (Sigma-Aldrich). Phorbol-12-myristate-13-acetate (PMA; Sigma-Aldrich) in the third tube served as a positive control. 5 µL of the specific substance was added to each tube. After the incubation times of 10 min and 20 min in a water bath at 37 °C, the reaction in all three tubes was stopped in a final step by cooling the samples to 4 °C and then adding the dye propidium iodide (PI, P1304MP; Thermo Fisher Scientific) for DNA staining. The final concentrations of the substances are shown in Table 7.

### 4.8. Flow Cytometric Measurement

After completing the incubation steps, cells from each prepared tube were analyzed using FACS Calibur and CellQuest Pro software version 5.2 (both BD Biosciences, Erembodegem, Belgium) at excitation wavelengths 488 nm and 633 nm. The forward and side scatter emission was used to determine the relative cell size (FSC) and the relative granularity (SSC). After detection, the cells were counted, as described in more detail in Section 4.11 and Section 4.12. The emission of the different wavelengths of the three dyes used (fluorescein isothiocyanate (FITC), phycoerythrin (PE) and allophycocyanin (APC) [67]) was determined.

### 4.9. Analysis of the Surface Epitope Tests

The data generated by the flow cytometer were analyzed using the FlowJo (Version 10.0.7 LLC, Ashland, OR, USA). Experiments with fluorescence intensities < 0 and measured cell numbers < 300 cells per tube were excluded. A total of 5000 cells were counted per tube.

Neutrophils were identified by their typical patterns displayed in forward-scatter (FSC) and side-scatter (SSC) light (Figure 10). For immunophenotyping, cell-surface expression of various antigens on PMNs was investigated using the following commercially available antibodies (see Appendix A), which were labeled with either phycoerythrin (PE), fluorescein isothiocyanate (FITC), or allophycocyanin (APC) according to the manufacturer’s protocol [68]. Surface epitope analysis was performed in the same way for each antibody used and was applied to both cPMNs (Figure 10a) and tPMNs (Figure 10b). The results were summarized as numerical values of the median of each point cloud and tabulated in Microsoft Excel (Microsoft Excel Version 2016 Microsoft Corporation, WA, USA).

### 4.10. Analysis of ROS Measurements

Analyses of the ROS experiments were also performed with Flow Jo in the same way as the surface epitope experiments (see Section 4.9). Once again, granulocytes were gated in SSC vs. FSC (Figure 11). By plotting the SSC against the fluorescence channel FL3; PI) (see Appendix A), living cells (PInegative) could be identified. Living cells were further analyzed to assess their ROS production (Figure 11). In FL1 vs. SSC, oxidized DHR 123 was visible as rhodamine-123. Results were summarized as the median intensity [MFI] of each cell population. Microsoft Excel was used to tabulate the percentage of avital cells and the median intensity of Rhodamine 123 as an indicator of ROS production. Each blood and aspirate sample was analyzed analogously (Figure 9).

### 4.11. Cell Counts in the Endotracheal Aspirate Using CASY

The cell counts in the endotracheal aspirate were analyzed using the “CASY cell counter” (Omni Live Science, Bremen, Germany). For this purpose, 20 µL of the aspirate was carefully mixed with 5 mL of prepared “Casyton-solution” (Omni live science, dilution 1:251). Subsequently, 3 × 400 µL were aspirated through the CASY for each measurement, and the mean value of the three measurements was calculated. All values > 8 µm were taken into account.

### 4.12. Cell Counts in Endotracheal Aspirate Using Flow Cytometry

After filtering and preparation, blood and secretion samples were analyzed for cell count using FACS (flow cytometer FACS Calibur BD Biosciences) in addition to the CASY measurement. The number of cells per 100 µL of filtered secretion and the total number of cells obtained and detected from the sample were determined. In addition, the number of cells contained per sample volume was calculated.

### 4.13. Evaluation of the Flow Cytometric Data

The generated data was evaluated using the FlowJo program (FlowJo Version 10.0.7 LLC, Ashland, OR, USA). The structured sample measurement processes through to statistical analysis can be seen in Figure 5 for the surface epitope tests and in Figure 6 for the ROS test. The generated data was documented and managed in tabular form using Microsoft Excel (Microsoft Excel Version 2016 Microsoft Corporation, Redmond, Washington, DC, USA). Further statistical analysis was performed using SPSS (SPSS Statistics Version 29 (IBM, Armonk, NY, USA).

### 4.14. Statistics

The statistical analysis was performed using SPSS Statistics (Version 29, IBM, Armonk, NY, USA). All variables were tested for normal distribution using Kolmogorov-Smirnov test. Given normal distribution and variance homogeneity, the mean values (MV) for a multiple comparison were tested for significant differences using one-way analysis of variance (ANOVA) with the specification of the standard deviation (±SD). In the case of variance inhomogeneity, the Kruskal–Wallis test was used to specify the median. The test for homogeneity of variance was performed using the Levene test. The subsequent post-hoc analysis for existing variance homogeneity was performed according to Bonferroni. The Dunnett T3 test was used for existing variance inhomogeneity. In the case of normal distribution, the distribution of the results is illustrated by bar charts with the mean ± 95% confidence interval, alternatively as a box plot showing median, lower quartile and calculated minima and maxima. Statistical outliers are shown as circles, and extreme values are asterisks. Blank values were subtracted beforehand. Cell numbers measured by flow cytometry were checked for plausibility (numbers < 300 per measurement were sorted out). An error probability of *p* < 0.05 was considered statistically significant.

## 5. Conclusions

This study revealed that CD66b expression on cPMNs from patients with ARDS was higher than on cPMNs from patients without ARDS and that CD66b expression on PMNs was higher in lung aspirate than in the blood. CD62L expression on tPMNs with ARDS was lower than on cPMNs without ARDS. Overall, CD62L was lower in aspirate than in blood. LOX-1 and FMLP-R were more highly expressed in PMNs in secretion than in blood. CD49d was higher in PMN secretory ARDS than in PMN secretory healthy, whereby CD29 showed a similar trend without significance. Excitability of ROS production was higher for cPMNs than for tPMNs from the secretion. In addition, the excitability of PMNs with ARDS was lower than without ARDS in the respective blood and aspirate groups. The results of this work suggest that PMNs have other altered functions and characteristics in ARDS.

In particular, altered levels of CD49d/CD29 (integrins that bind collagen III, among others), combined with increased ROS activity when PMNs come into contact with type III collagen, represent an interesting association. The results of this work, in conjunction with results from previous publications [24,26] support the hypothesis that the interaction of PMNs with type III collagen via interaction with CD49d/CD29 may play an important role in the promotion of ARDS. The investigation of integrin-mediated cell-matrix interactions not only represents a valuable area of basic research but also serves even today as a crucial therapeutic pillar in the treatment of inflammatory diseases. Monoclonal antibodies such as Natalizumab for multiple sclerosis and Vedolizumab for chronic inflammatory bowel diseases are already employed to inhibit integrin-mediated interactions of lymphocytes with their environment. Therefore, it is mandatory that future studies also investigate the interactions of polymorphonuclear leukocytes (PMNs) with their surroundings to elucidate potential integrin involvement, thereby advancing therapy-oriented research. Influencing a possible type III collagen—PMN (activating) interaction could prove useful in advanced ARDS treatment. Anyway, research in this direction should be pursued [60].

## Figures and Tables

**Figure 1 ijms-25-12547-f001:**
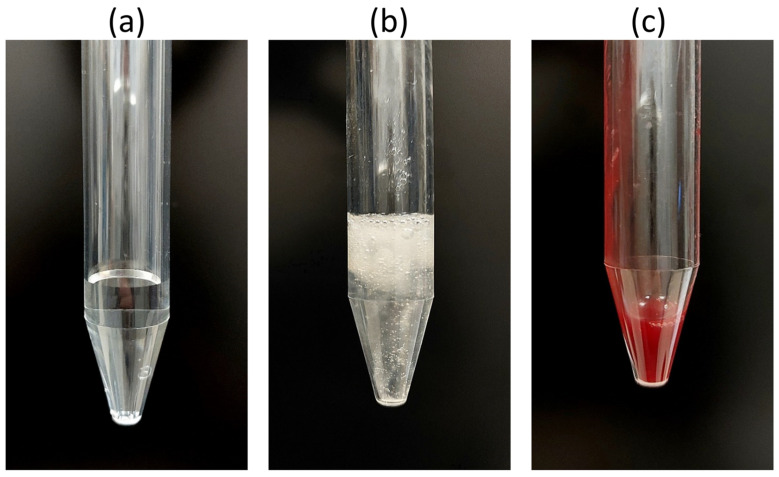
(**a**) Serous endotracheal aspirate (**b**) Mucuous endotracheal aspirate (**c**) Sanguineous endotracheal aspirate.

**Figure 2 ijms-25-12547-f002:**
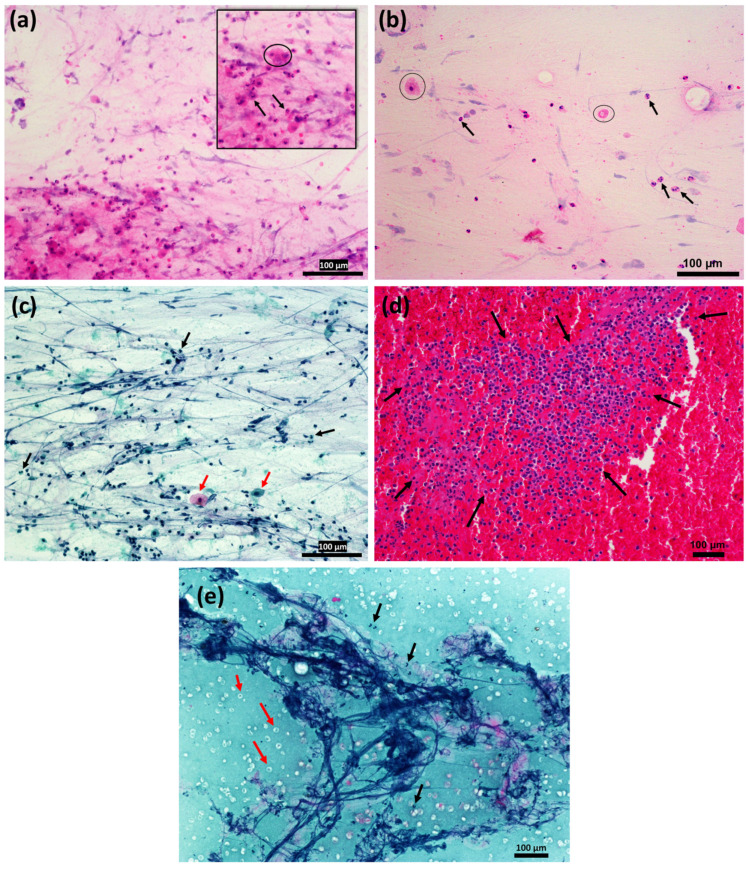
Cytological staining of endotracheal aspirate. (**a**) Hematoxylin-eosin staining of mucosal endotracheal aspirate: Black arrows = rod-nucleated and segment-nucleated PMNs, Black circle = alveolar macrophage. The varying intensity of the pink coloration is due to the mucous consistency of the endotracheal aspirate, resulting in intensely pink-colored areas in the lower left edge of the image and lighter areas in the upper right area of the image. Purple threads of mucus can also be seen. (**b**) HE staining of mucosal endotracheal aspirate. Black arrows = rod-nucleated and segment-nucleated PMNs, Black circles = alveolar macrophages, probably with phagocytosed. The loosened and less cell-dense appearance of this image is due to the staining technique. (**c**) Papanicolaou stain of mucosal endotracheal aspirate. Respiratory epithelia may also be present among the cells (Red arrows = squamous epithelial cells). There are also dark cell nuclei, e.g., of PMNs (black arrows), as well as numerous dark blue mucus threads interspersing the image. (**d**) HE staining of a sanguineous endotracheal aspirate. PMNs colored in dark purple (black arrows) dominate the aspirate next to the pink-colored blood. Macrophages and plasma cells are also present between the PMNs. (**e**) Papanicolaou staining of sanguineous endotracheal aspirate. Mucus threads can be seen in strong dark blue. Red arrows mar brightly colored erythrocytes. PMNs are marked with black arrows.

**Figure 3 ijms-25-12547-f003:**
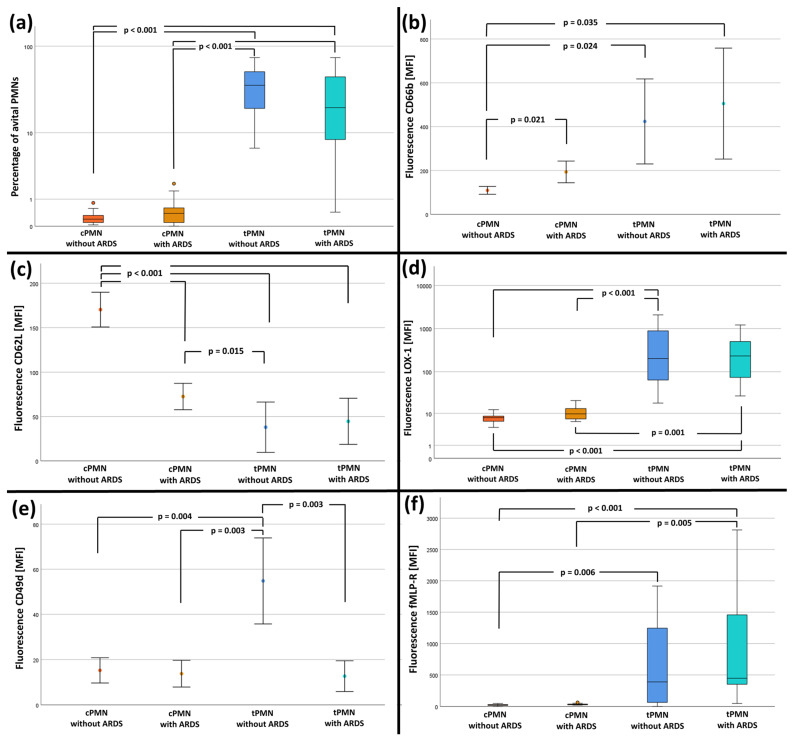
Results of flow cytometry measurements Data are shown as median ± interquartile ranges or mean ± 95% confidence interval and *p*-values. (**a**) Comparison of the percentage of avital PMN, (**b**) Comparison of MFI(CD66b), (**c**) Comparison of MFI(CD62L). (**d**) Comparison of MFI(LOX-1), (**e**) Comparison of MFI(CD49d), (**f**) Comparison of MFI(fMLP-R).

**Figure 4 ijms-25-12547-f004:**
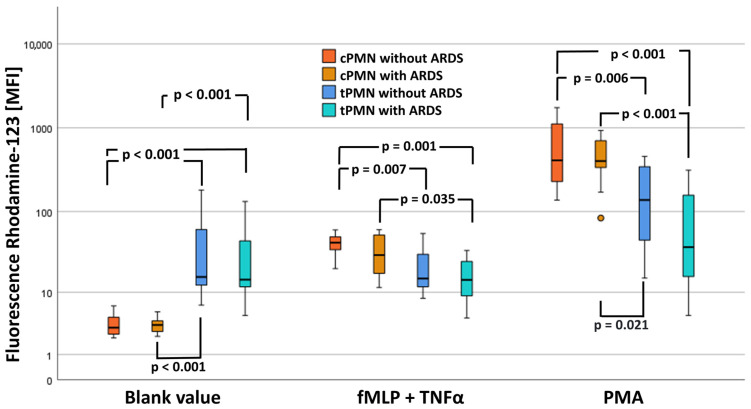
Comparison of MFI(DHR) subdivided into activating substances and study groups. Data are shown as median ± interquartile ranges and *p*-values.

**Figure 5 ijms-25-12547-f005:**
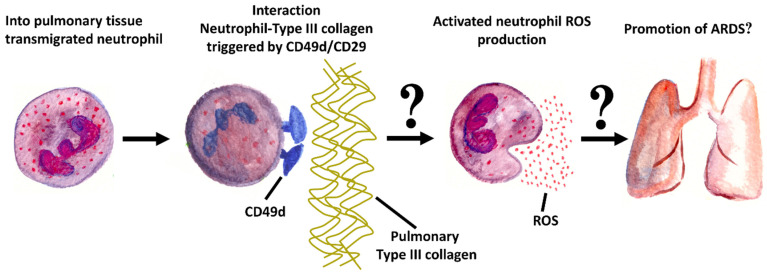
The results of this work, in conjunction with results from previous publications [24,26], support hypotheses that the interaction of PMNs with type III collagen via interaction with CD49d/CD29 may play an important role in the promotion of ARDS.

**Figure 6 ijms-25-12547-f006:**
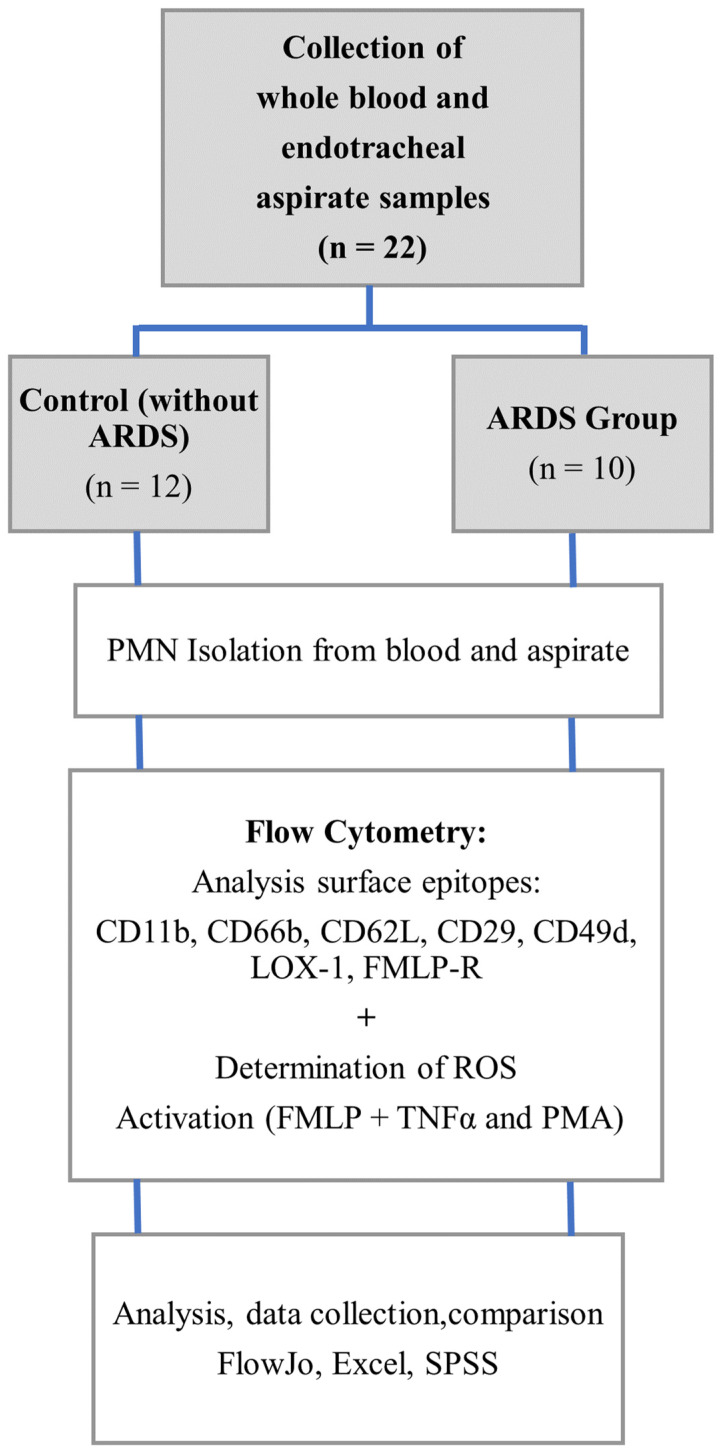
Workflow of the experiments.

**Figure 7 ijms-25-12547-f007:**
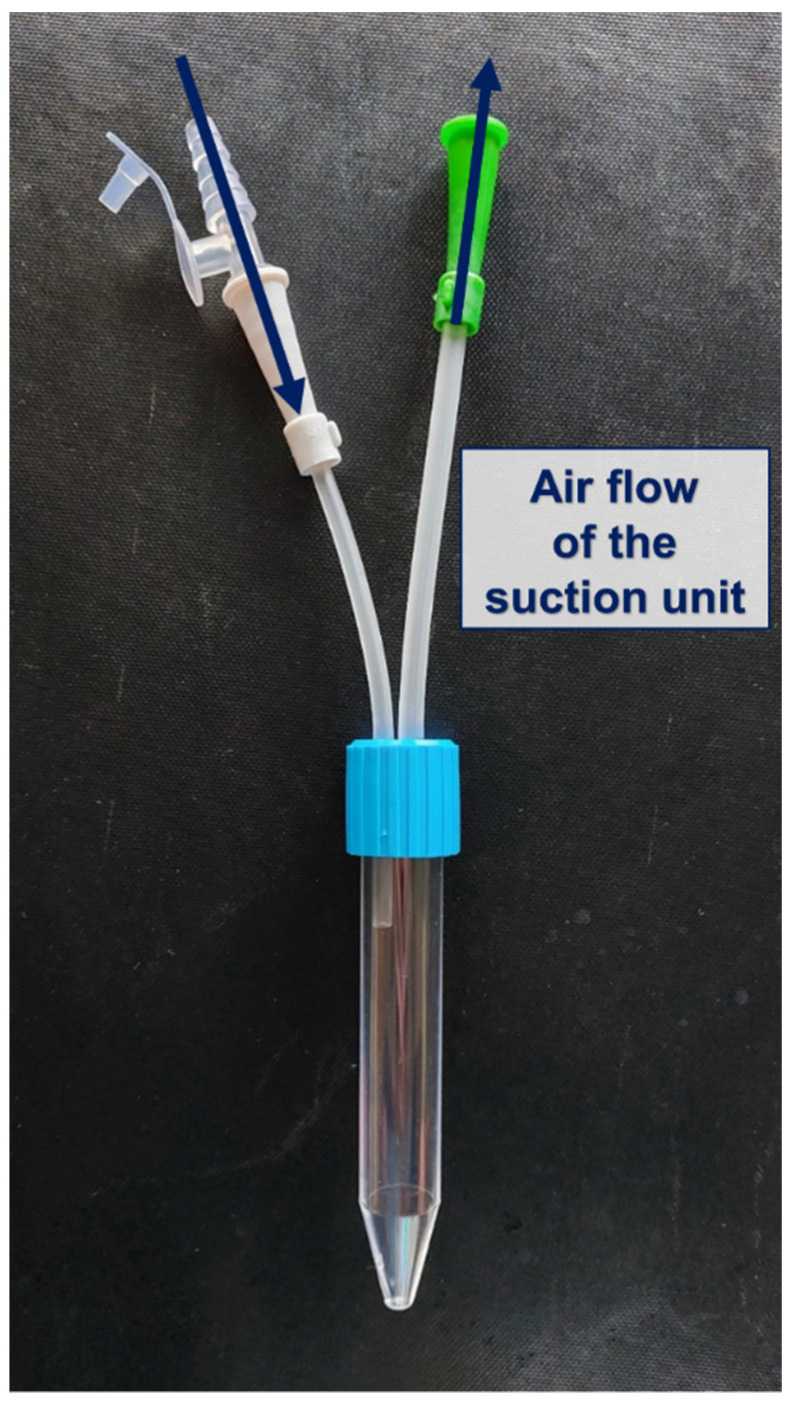
Illustration of the tracheal suction set. The dark blue arrows illustrate the direction of the airflow.

**Figure 8 ijms-25-12547-f008:**
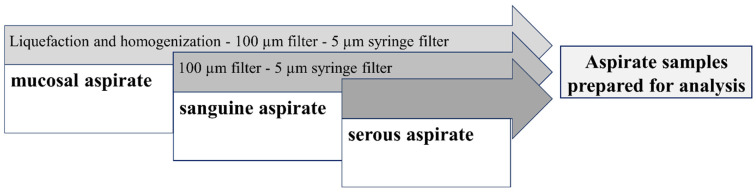
Preparation methods for the aspirate depend on the original quality of the tPMN isolation.

**Figure 9 ijms-25-12547-f009:**
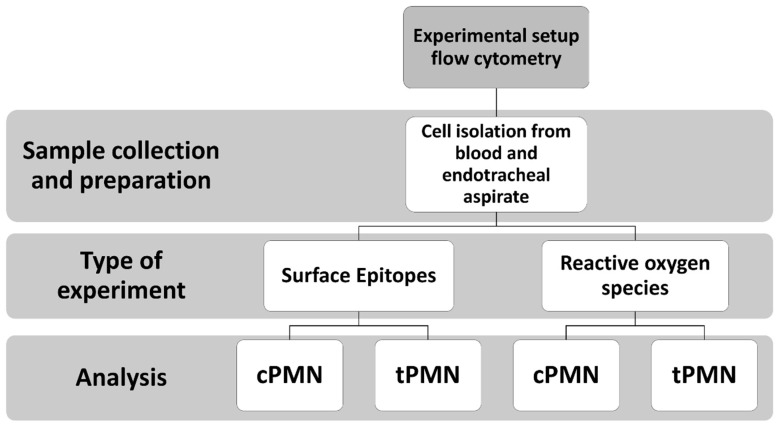
Experimental setup for flow cytometric analysis.

**Figure 10 ijms-25-12547-f010:**
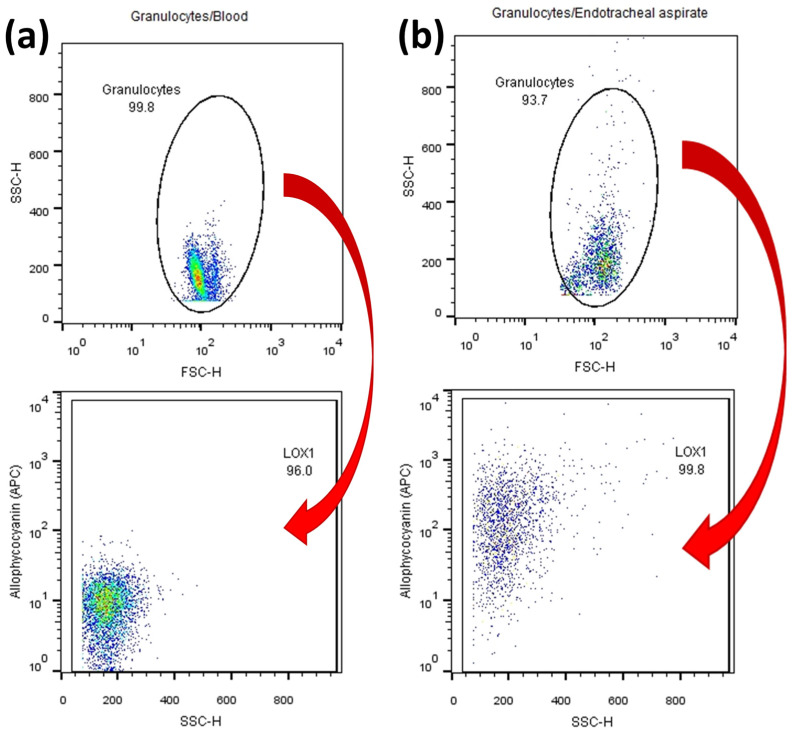
(**a**) Analysis of surface epitopes of (**a**) cPMNs or (**b**) tPMNs using APC-conjugated antibody against LOX-1. Abbreviations: Granulocytes = PMNs; SSC-H = side scatter height; FSC-H = forward scatter height.

**Figure 11 ijms-25-12547-f011:**
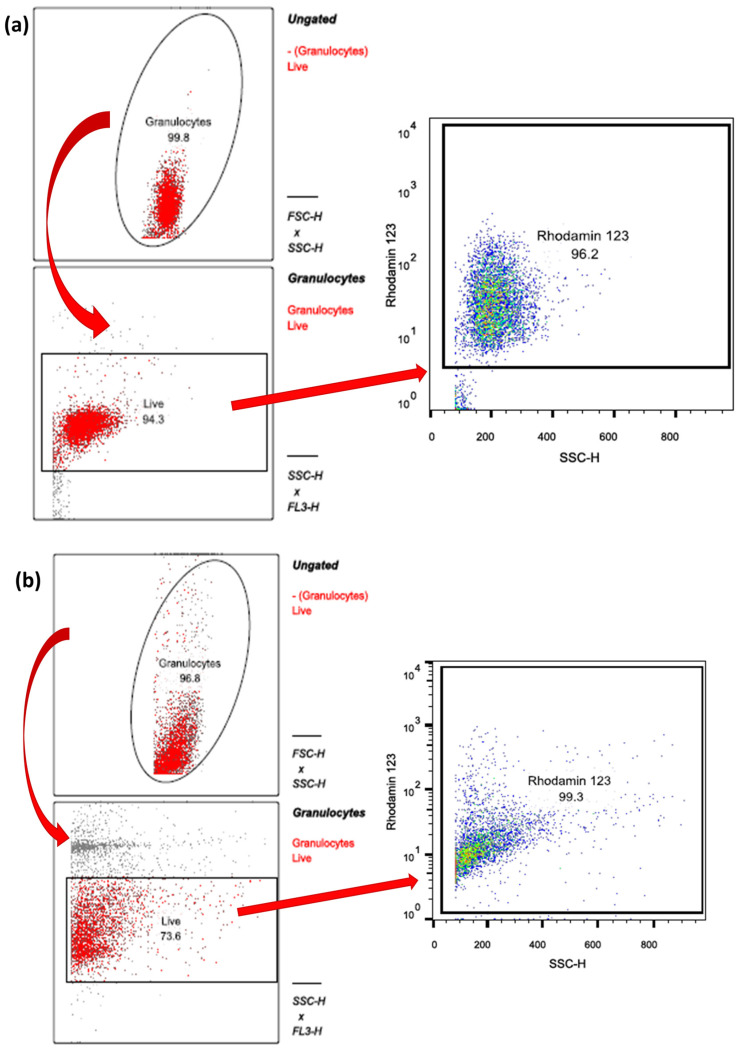
(**a**) Analysis of (**a**) cPMN and (**b**) tPMN ROS production (Red arrows illustrate the further procession of selected cells. Abbreviations: Granulocytes = PMNs; SSC-H = side scatter height; FSC-H = forward scatter height; FL3-H = fluorescence 3; Live = living cells.

**Table 1 ijms-25-12547-t001:** Overview of patients’ personal data in the control group (without ARDS).

Number of Study Participants Without ARDS	*n* = 12
Median age	65.5 years
Age Range	38–76 years
Sex	*n* = 3 female, *n* = 9 male
VV-ECMO	*n* = 0
VA-ECMO	*n* = 2
Dialysis	*n* = 0

**Table 2 ijms-25-12547-t002:** Overview of patients’ personal data in the ARDS group.

Number of Study Participants: Patient Group (with ARDS)	*n* = 10
Median Age	54.5 years
Age Range	44–84 years
Sex	*n* = 7 female, *n* = 3 male
VV-ECMO	*n* = 3
VA-ECMO	*n* = 0
Dialysis	*n* = 4

**Table 3 ijms-25-12547-t003:** Comparison of cell measurements using CASY and flow cytometry.

CASY	HomogenizationFilteringPipetting LossCell Isolation	Flow Cytometry
Native sampleEndotracheal aspirate	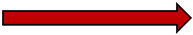	Prepared SampleAspirate
Cell content: 262/µL	>	Cell content:11/µL

**Table 4 ijms-25-12547-t004:** Overview of fluorescently labeled antibodies. Abbreviations: phycoerythrin (PE), fluorescein isothiocyanate (FITC), allophycocyanin (APC).

Surface Epitopes	Coupled Dye	Product and Manufacturer
CD11b	PE	PE anti-human CD11b Clone: ICRF44 150 µg/mL BioLegend, San Diego, CA, USA
CD62L	FITC	FITC anti-human CD62L Clone: DREG-56 BioLegend, San Diego, CA, USA
CD66b	APC	APC anti-human CD66b Clone: G10F5 200 µg/mL BioLegend, San Diego, CA, USA
LOX-1	APC	APC Anti-human, LOX-1 Antibody Clone: REA1188 Miltenyl Biotec, Bergisch Gladbach, Deutschland
CD49d	PE	PE anti-human alpha 4/CD49d Clone: #7.2R Biotechne, Minneapolis, MN, USA
CD29	FITC	FITC anti-human integrin beta 1/CD29 Clone: TS2/16 Thermo Fisher Scientific, Waltham, MA, USA
fMLP-Receptor	APC	APC Anti-human, fMLP receptor Antibody Clone: REA169 Miltenyl Biotec, Bergisch Gladbach, Deutschland

**Table 5 ijms-25-12547-t005:** The experimental procedure involves surface epitope analysis (BV = blank value; X = test step is performed, AB = antibody).

One Preparation Each for Blood and Secretion	Blank Value	CD11b (PE) + CD62L (FITC) + CD66b (APC)	LOX-1 (APC)	CD49d (PE) + CD29 (FITC) + FMLP-R (APC)
1. PMNs	20 µL PMNs of blood /100 µL PMNs of aspirate	20 µL PMNs of blood /100 µL PMNs of aspirate	20 µL PMNs of blood /100 µL PMNs of aspirate	20 µL PMNs of blood /100 µL PMNs of aspirate
2. PBS (+CaMg, 4 °C) to isolated PMNs from blood	50 µL	50 µL	50 µL	50 µL
3. 5 µL AB each	-	5 µL AB each	5 µL AB each	5 µL AB each
4. Incubate (15 min in the dark at 4 °C)	X	X	X	X
5. +PBS	2 mL	2 mL	2 mL	2 mL
6. Centrifuge at 4 °C for 3 min and 425 g	X	X	X	X
7. PBS (+CaMg, 4 °C + darkness)	250 µL PBS	250 µL PBS	250 µL PBS	250 µL PBS
8. Fluorescent measurements	X	X	X	X

**Table 6 ijms-25-12547-t006:** Experimental setup for flow cytometric ROS measurement.

One Preparation Each for Blood and Aspirate	Blank Value	TNFα + FMLP	PMA
1. PBS (4 °C)	500 µL (PMNs of blood) /420 µL (PMNs of aspirate)	500 µL (PMNs of blood) /420 µL (PMNs of aspirate)	500 µL (PMNs of blood) /420 µL (PMNs of aspirate)
2. PMNs	20 µL (PMNs of blood) /100 µL (PMNs of aspirate)	20 µL (PMNs of blood) /100 µL (PMNs of aspirate)	20 µL (PMNs of blood) /100 µL (PMNs of aspirate)
3. +DHR	5 µL	5 µL	5 µL
4. +SNARF	5 µL	5 µL	5 µL
5. +TNFα	-	5 µL	-
6. Incubate for 10 min at 37 °C	X	X	X
7. +FMLP	-	5 µL	-
8. +PMA	-	-	5 µL
8. Incubate for 20 min at 37 °C	X	X	X
9. 4 °C until measurement, +PI	5 µL	5 µL	5 µL
10. Flow cytometric measurement	X	X	X

**Table 7 ijms-25-12547-t007:** Used substances in their final concentrations.

Substance	Final Concentration
DHR	1 µm
SNARF	100 nM
TNFα	15 nM
FMLP	100 nM
PMA	100 nM
PI	15 µM

## Data Availability

The data presented in this work are available on request from the corresponding author.

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
