# Peer review of "Neutrophils in the Spotlight—An Analysis of Neutrophil Function and Phenotype in ARDS"

_ijms, 2024, doi:10.3390/ijms252312547_

Round 1

Reviewer 1 Report

Comments and Suggestions for Authors

This is a very interesting original research article about the immunophenotypic and functional characteristics of circulating and airway neutrophils and their differences between patients with or without acute respiratory distress syndrome (ARDS). The rationale is adequately delineated, the methodology is well-structured, and the results are clearly presented. Uncovering the particular alterations of neutrophils and unraveling their involvement in ARDS is of great clinical importance, as it facilitates the better understanding of the pathogenesis of the condition and their consideration as potential disease biomarkers and therapeutic targets. A few minor comments are as follows:

1. The findings regarding the differences in the expression of fMLP-receptor between circulating and tracheal neutrophils (figure 18) could be briefly mentioned in the text.

2. The conclusions could be elaborated by commenting on the potential clinical importance of the findings and the future perspectives of translational research on neutrophils and neutrophilic inflammation in ARDS.

3. Figures 7-12 could be unified into a single figure with the multiple images of the macroscopic properties and the cytological assays of the aspirates. Similarly, figures 13-18 could be unified into a single figure with the multiple graphs regarding the immunophenotypic differences of neutrophils between the study groups.

4. "CD11b" should be replaced by "CD62L" in the heading of section 3.2.2 (The heading "Interpretation of CD11b expression in ARDS" appears in both 3.2.1 and 3.2.2 sections).

5. The whole manuscript should be screened for minor typos.

Reviewer 2 Report

Comments and Suggestions for Authors

The subject of the manuscript is very interesting, however I think it needs many improvements before publication, both in the drafting and in the experiments conducted.

I advise the authors to better explain the results obtained, and not simply put the reference to the figure/table. I also think that only the figure or table of the same result is sufficient, and in the case of the tables i recommend as much as possible to unite them by also including the significance data.

I also recommend better defining the study population by adding some more clinical data.

As for the experimental part, I recommend the authors to confirm the cytofluorometry data with in vitro or molecular data to support the discussion.

Best regards

Reviewer 3 Report

Comments and Suggestions for Authors

The manuscript is interesting and quite well written, I have some suggestions:

1) Abstract. Overall, PMNs appear to be in a more activated state in lung secretions than in blood, as indicated 30 by higher CD66b and lower CD62L expression, higher constitutive ROS production and lower ex-31 citability with fMLP and TNFα. In context of possible CD49d-triggered ROS production, it is note-32 worthy that CD49d is downregulated in secretion from patients with ARDS compared to patients 33 without. This phenotypic and functional PMN-characterization can provide valuable diagnostic 34 and therapeutic information for the intensive care treatment of ARDS-patients. 35 I think that abstract might be beneficial to include a sentence that briefly summarizes the key findings of the study. This can provide readers with a quick overview of the research. 

2) Keywords: I suggest to insert some "Keywords". I think that also keywords can provide readers with a quick overview of the research. 

3) 1. Introduction 38. 1.1. Neutrophil Involvement in the Pathophysiology of Acute Respiratory Distress 39. ARDS is a clinical disease pattern associated with diffuse pulmonary inflammation 40 and edema leading to hypoxemia. Global awareness of ARDS was heightened by the 41 COVID-19 pandemic in 2020 with a sharp rise in ARDS incidence. ARDS can be triggered 42 by a variety of infectious or non-infectious causes, such as pneumonia or extra-pulmonary. Authors are kindly requested to emphasize the current concepts about these issues in the context of recent knowledge and the available literature. These articles should be quoted in the References list. References 1. Novel Therapeutic Target Critical for SARS-CoV-2 Infectivity and Induction of the Cytokine Release Syndrome. Cells. 2023 May 7;12(9):1332. doi: 10.3390/cells12091332. PMID: 37174732; PMCID: PMC10177205.

2. Cytokine Profiles as Potential Prognostic and Therapeutic Markers in SARS-CoV-2-Induced ARDS. J Clin Med. 2022 May 24;11(11):2951. doi: 10.3390/jcm11112951. PMID: 35683340; PMCID: PMC9180983.

3. Immune Response Dynamics and Biomarkers in COVID-19 Patients. Int. J. Mol. Sci. 202425, 6427. https://doi.org/10.3390/ijms25126427

4) In this way, the investigation of expression levels of these surface epitopes not only 96 enables further elucidation of ARDS pathophysiology, but may also help to discover tar-97 gets for new pharmacological therapies. To investigate a possible connection in more de-98 tail, we carried out cytological examinations and immunophenotyping in patients with 99 and without ARDS, and also examined the ROS production of PMNs from blood and 100 tracheal secretions. I suggest to underline the novelty of the study.

5) 2. Results. I suggest to underline the the most important results to clarify the data.

6) 3. Discussion 207

3.1. Interpretation of the macroscopic properties and cytological staining of the endotracheal 208 aspirate 209 Macroscopic assessment of the endotracheal aspirate provides information on the 210 physiological state of the airways and allows conclusions about potential changes in the 211 course of ventilator-dependent lung diseases. Serous transparency of the aspirate (see Fig-212 ure 7a) shows no admixture of blood or mucus. These clear characteristics indicate a non-213 pathological composition of the aspirate with a higher liquid content and a low content of 214 solids. Since this appearance was particularly present in the control group without ARDS, 215 it can serve as a reference point for comparison with the other macroscopic appearances 216 “mucuous” (see Figure 7b) and “sanguineous” (see Figure 7c) in presence of ARDS. Se-217 rous aspirate indicates a “more normal” secretion and absence of pathological changes 218 [34]. The discussion section needs to be improved.  It is necessary to clarify the results obtained and compare them with previous or similar studies. 

7) I suggest also to underline the limitations of the study.

8) 5. Conclusions 599

This study revealed that CD66b expression on cPMNs from patients with ARDS was 600 higher than on cPMNs from patients without ARDS and that CD66b expression on PMNs 601 was higher in lung aspirate than in the blood. CD62L expression on tPMNs with ARDS 602 was lower than on cPMNs without ARDS. Overall, CD62L was lower in aspirate than in 603 blood. LOX-1 and FMLP-R were more highly expressed on PMNs in secretion than in 604 blood. CD49d was higher in PMN secretory ARDS than in PMN secretory healthy, 605 whereby CD29 showed a similar trend without significance. Excitability of ROS produc-606 tion was higher for cPMNs than for tPMNs from the secretion. In addition, excitability of 607 PMNs with ARDS was lower than without ARDS in the respective blood and aspirate 608 groups. The results of this work suggest that there are other altered functions and charac-609 teristics of PMNs in ARDS. I suggest to underline the novelty of the study and the future prospects.

Round 2

Reviewer 2 Report

Comments and Suggestions for Authors

I think the manuscript has been sufficiently implemented by the authors with the requested suggestions

Reviewer 3 Report

Comments and Suggestions for Authors The authors adequately answered my questions. They edited the manuscript and took my suggestions into account. In my opinion this improved the manuscript. I have no further comments.